# Attrition in serum anti-DENV antibodies correlates with high anti-SARS-CoV-2 IgG levels and low DENV positivity in mosquito vectors—Findings from a state-wide cluster-randomized community-based study in Tamil Nadu, India

Sivaprakasam T. Selvavinayagam[1], Sathish Sankar[2], Yean K. Yong[3], Abdul R. Anshad[4], Samudi Chandramathi[5], Anavarathan Somasundaram[6], Sampath Palani[1], Parthipan Kumarasamy[1], Roshini Azhaguvel[1], Ajith B. Kumar[1], Sudharshini Subramaniam[6], Manickam Malathi[7], Venkatachalam Vijayalakshmi[7], Manivannan Rajeshkumar[1], Anandhazhvar Kumaresan[1], Ramendra P. Pandey[8], Nagarajan Muruganandam[9], Natarajan Gopalan[10], Meganathan Kannan[11], Amudhan Murugesan[12], Pachamuthu Balakrishnan[13], Siddappa N. Byrareddy[14], Aditya P. Dash[15], Vijayakumar Velu[16], Marie Larsson[17], Esaki M. Shankar[4]*, Sivadoss Raju[1]*

1 State Public Health Laboratory, Directorate of Public Health and Preventive Medicine, DMS Campus, Teynampet, Chennai, Tamil Nadu, India, 2 Department of Microbiology, Saveetha Dental College and Hospitals, Saveetha Institute of Medical and Technical Sciences, Centre for Infectious Diseases, Saveetha University, Chennai, Tamil Nadu, India, 3 Laboratory Centre, Xiamen University Malaysia, Sepang, Selangor, Malaysia, 4 Department of Biotechnology, Infection and Inflammation, Central University of Tamil Nadu, Thiruvarur, India, 5 Faculty of Medicine, Department of Medical Microbiology, University of Malaya, Lembah Pantai, Kuala Lumpur, Malaysia, 6 Institute of Community Medicine, Madras Medical College, Chennai, Tamil Nadu, India, 7 Institute of Vector Control and Zoonoses, Hosur, Tamil Nadu, India, 8 School of Health Sciences and Technology, UPES, Dehradun, Uttarakhand, India, 9 Regional Medical Research Centre, Indian Council of Medical Research, Port Blair, Andaman and Nicobar Islands, India, 10 Department of Epidemiology and Public Health, Central University of Tamil Nadu, Thiruvarur, India, 11 Department of Biotechnology, Blood and Vascular Biology, Central University of Tamil Nadu, Thiruvarur, India, 12 Department of Microbiology, Government Theni Medical College and Hospital, Theni, Tamil Nadu, India, 13 Saveetha Institute of Medical and Technical Sciences, Center for Infectious Diseases, Saveetha Medical College and Hospital, Saveetha University, Chennai, Tamil Nadu, India, 14 Department of Pharmacology and Experimental Neuroscience, University of Nebraska Medical Center, Omaha, Nebraska, United States of America, 15 Asian Institute of Public Health University, Bhubaneswar, Odisha, India, 16 Department of Pathology and Laboratory Medicine, Division of Microbiology and Immunology, Emory University School of Medicine, Emory National Primate Research Center, Emory Vaccine Center, Atlanta, Georgia, United States of America, 17 Department of Biomedical and Clinical Sciences, Division of Molecular Medicine and Virology, Linköping University, Linköping, Sweden

055 These authors contributed equally to this work.
* sivraju@gmail.com (SR); shankarem@cutn.ac.in (EMS)

## Abstract

The decline in dengue incidence and/or prevalence during the COVID-19 pandemic (2020–22) appears to be attributed to reduced treatment-seeking rates, under-reporting, misdiagnosis, disrupted health services and reduced exposure to mosquito vectors due to prevailing lockdowns. There is limited scientific data on dengue virus (DENV) disease during the

**Data Availability Statement:** The authors confirm that the all data supporting the findings of this study are available within the paper and its Supplementary Information. Other data including de-identified participant data and specific datasets that have sensitive information which could compromise the privacy of research participants will be available upon reasonable request due to restrictions imposed by the institutional ethical committee. The data will be available beginning five months and ending three years after publication. Data requestor will be required to sign a data access agreement before data are released. Data requests can be sent to the Institutional Ethics Committee at dphethicscommittee@gmail.com.

**Funding:** The study was funded by the National Health Mission (https://www.nhm.tn.gov.in/en), Tamil Nadu (680/NGS/NHMTNMSC/ENGG/2021) for the Directorate of Public Health and Preventive Medicine to S.T.S. and S.R. M.L. is supported by grants through AI52731, the Swedish Research Council (https://www.vr.se/english.html), the Swedish, Physicians against AIDS Research Foundation, the Swedish International Development Cooperation Agency, SIDASARC, VINNMER for Vinnova, Linköping University Hospital Research Fund, CALF, and the Swedish Society of Medicine. V.V. is supported by the Office of Research Infrastructure Programs (ORIP/NIH) (https://orip.nih.gov/home) base grant P51 OD011132 to ENPRC. A.M. is supported by Grant No. 12020/04/2018, Department of Health Research, Government of India (https://dhr.gov.in/). The funders of the study had no role in the study design, data collection, data analysis, data interpretation, or writing of the report.

**Competing interests:** The authors have declared that no competing interests exist.

COVID-19 pandemic. Here, we conducted a community-based, cross-sectional, cluster-randomized survey to assess anti-DENV and anti-SARS-CoV-2 seroprevalence, and also estimated the spatial distribution of DENV-positive aedine mosquito vectors during the COVID-19 pandemic across all the 38 districts of Tamil Nadu, India. Using real-time PCR, the prevalence of DENV in mosquito pools during 2021 was analyzed and compared with the previous and following years of vector surveillance, and correlated with anti-DENV IgM and IgG levels in the population. Results implicate that both anti-DENV IgM and IgG seroprevalence and DENV positivity in mosquito pools were reduced across all the districts. A total of 13464 mosquito pools and 5577 human serum samples from 186 clusters were collected. Of these, 3.76% of the mosquito pools were positive for DENV. In the human sera, 4.12% were positive for anti-DENV IgM and 6.4% for anti-DENV IgG. While the anti-SARS-CoV-2 levels significantly correlated with overall DENV seropositivity, COVID-19 vaccination status significantly correlated with anti-DENV IgM levels. The study indicates a profound impact of anti-SARS-CoV-2 levels on DENV-positive mosquito pools and seropositivity. Continuous monitoring of anti-DENV antibody levels, especially with the evolving variants of SARS-CoV-2 and the surge in COVID-19 cases will shed light on the distribution, transmission and therapeutic attributes of DENV infection.

## Introduction

Dengue represents a global arboviral public health threat, caused by four serotypes of dengue virus (DENV1-4). *Aedes* mosquitoes (*Aedes aegypti* and *Aedes albopictus*), act as vectors in the tropical and subtropical regions making the disease hyperendemic across Asia and South America, Africa, the Middle East [1,2] and other temperate parts of the world [3]. The single-stranded positive-sense RNA-laden Flavivirus causes frequent concurrent epidemics involving different serotypes. While DENV2 appears to be associated with severe disease, there is evidence for the distribution of all DENV serotypes in Asia [4]. Dengue is classified as primary and secondary based on IgM:IgG ratio, and two types, viz. dengue without warning signs (DWWS) and dengue with warning signs (DWS) based on clinical manifestations [5,6]. The prognosis of dengue is determined by antibody-dependent enhancement (ADE), viral dynamics, and pre-existing antibody titers [7]. However, protean clinical manifestations, serotype heterogeneity, and co-infections pose a substantial challenge to patient management.

There is a growing interest in prevailing infections post the SARS-CoV-2 pandemic. As with other infections, there has been a shift in the dengue fever trend in 2020–22, when COVID-19 was witnessing a surge. Others have reported a 16–97% decrease in dengue cases during the COVID-19 pandemic [8–10]. Besides, there have also been reports of concomitant dengue together with other infectious agents, including SARS-CoV-2 [11–13]. The pressure that prevailed during the COVID-19 pandemic raised concerns over the lack of attention to dengue diagnosis, reduced treatment-seeking rates, potential for misdiagnosis, reduced availability of laboratory testing for dengue, and negative impact of lockdowns [10]. There has been a declining trend in dengue post-COVID-19 following an upsurge in 2019 [14], likely due to the global imposition of lockdowns [15]. In India, dengue incidence was reported to be ~188,000 (2017), 101,192 (2018) and 157,315 (2019) cases. However, the frequency of dengue declined abruptly to 45,585 (71%) (https://ncvbdc.mohfw.gov.in) [16].

Studies reporting dengue decline during the pandemic were often based on serological investigations (NS1/IgM/IgG). Our state-wide entomological surveillance and vector control

data indicated a significant reduction of DENV-positive mosquito pools in 2020 that remained low until 2023. This further substantiated our assumption of reduced DENV transmission due to lockdowns. We hypothesized that there is a correlation trend between the anti-SARS-CoV-2 IgG as well as anti-DENV IgM and IgG titers. Possibly, antibodies to SARS-CoV-2 could hinder the circulation of DENV either by cross-protection, antigenic mimicry or by masked effects of ADE [17]. The cross-reactive nature of anti-SARS-CoV-2 has been reported against certain antigens and vaccines [18]. Antibodies to spike and receptor-binding domain (S1-RBD) have been shown to cross-react with both DENV envelope protein (E) and non-structural protein 1 (NS1) in experimental animals [19].

Constant monitoring of disease prevalence and entomological surveillance together with risk factors of viral transmission are critical for highly endemic countries like India. Here, we conducted a community-based, cross-sectional, cluster-randomized survey to assess the sero-prevalence of dengue and SARS-CoV-2, and correlated with DENV positivity in aedine mosquito vectors in Tamil Nadu, India in December 2021. The primary and secondary DENV infections along with the antibody titers were correlated with the SARS-CoV-2 IgG in the population.

## Methods

### Ethical approval

This study was performed within the ethical standards of the Declaration of Helsinki. The procedures and/or study protocols were reviewed and approved by the Institutional Ethical Committee of the Madras Medical College (Approval No. 03092021 and 34122021), Chennai, India. All participants were aged >18 years and provided written informed consent to participate in the investigation. The start date of the participant recruitment was 22nd December 2021 and the end date was 19th March 2023.

### Inclusivity in global research

Additional information regarding the ethical, cultural, and scientific considerations specific to inclusivity in global research is included in the Supporting Information (**S1 Checklist**).

### Mosquito sampling

The eggs, larvae, and adult Aedes mosquitoes were collected from all 38 districts of Tamil Nadu indoors and outdoors. The sampling and testing have been carried out as part of the routine surveillance program since 2016 for the prevention and control of vector-borne diseases by the Department of Public Health, Tamil Nadu, India. Here, we compared and analyzed the samples collected from 2016 to mid-2024 for possible correlation with the seroprevalence of SARS-CoV-2 and DENV. The adult female Aedes mosquitoes captured were identified and isolated using a standard morphological method [20] and were transported to the processing laboratory. The eggs hatched after an incubation period of 15 days at the Regional Entomology Laboratory. The larvae and adults were identified and the dried adult mosquito samples were transported in zip-lock covers or microcentrifuge tubes to the State Public Health Laboratory (SPHL), Chennai, and the Institute of Vector Control and Zoonosis (IVCZ), Hosur, India.

### Sample processing

Engorged adult female mosquito pools (n = 25) collected from specific trap areas were prepared. The dried adult mosquito pools were crushed and homogenized with 200 µl of Leibovitz's media (L-15) twice with a Teflon pestle homogenizer before centrifuging at 1000 rpm,

4°C for 10 minutes. The supernatant was aliquoted in tubes and stored at −80°C until further use.

### RNA extraction and molecular detection of DENV

The viral RNA from the homogenized mosquito supernatant was extracted using HiPurA prefilled medium plates-T kit (HiMedia, Maharashtra, India) using a KingFisher Flex automated extraction system (Thermo Fisher Scientific, Waltham, USA). The mosquito pools were screened for DENV using a DENV real-time reverse transcriptase PCR kit (Helini Biomolecules, Chennai, India) in the Quant Studio 5 Real-time PCR System (Applied Biosystems, Waltham, USA) according to the manufacturer's instructions. The kit contained pan-DENV-specific primers and probes for the quantification of DENV1-4 in the FAM channel. The target sequence 5'UTR is highly conserved across all DENV serotypes. The linear range of the assay kit ranged from 1 to $1 \times 10^9$ copies/µl. Possible PCR inhibition and RNA purification efficiency were controlled using an internal amplification control in the HEX channel. In the RT-PCR assay, the cycle threshold (Ct) cut-off value range for DENV positivity was between 13 and 35. Any Ct value >35 was considered DENV-negative while a value <13 was diluted and the assay was repeated.

### Study design and participants

A community-based, cross-sectional, randomized cluster sampling was carried out to assess the seroprevalence of dengue in all 38 districts of Tamil Nadu, India. All the participants were aged >10 years, and accented/consented to involve in the investigation. The individuals also included those with suspected or confirmed past dengue infection. From the 38 states, a total of 186 clusters were selected using stratified random sampling.

The size of the cluster was determined based on the population-to-size ratio and was considered an adequate representation of the state. After identifying the cluster, the houses within the cluster were marked and numbered. During the study, a random household was selected and considered as the first household for the study and at least 30 to the left of the primary house were included in the study. The survey team collected all the identification details of the members including socio-demographic details from the selected household. From each household, one respondent was randomly identified for survey sampling using the Kish grid method [21]. Participants were also given a unique ID for identification. At the time of sampling, the participants were enquired about dengue and COVID-19 status, vaccination status, and the type of SARS-CoV-2 vaccine administered.

### Clinical specimens

Considering a 76.9% dengue seroprevalence, a design effect of two, a confidence level of 95% and a precision value of 1.3, the required sample size was calculated as 20. Assuming one-third of the randomly assigned sample would become ineligible due to hemolysis during transportation and refusal to participate in the study, the final sample size was established as 30 per cluster. Two millilitres of venous blood were collected for serum separation before transporting to the District Public Health Laboratory for dengue IgM and IgG ELISA. The other aliquot was transported to the State Public Health Laboratory for SARS-CoV-2 IgG assay.

### Anti-DENV IgM and IgG

The extracted serum was tested for IgM as well as IgG antibodies using Panbio Dengue IgM and IgG capture ELISA (Abbot Diagnostics, South Korea). Cut-off values were determined as

per the manufacturers' instructions. Panbio Units (PU) were calculated as 10 times the value of sample absorbance divided by the cut-off value. A PU value of >11 and <9 was considered positive and negative, respectively. Any value between 9 and 11 was considered equivocal and was tested with the same assay and considered negative if the repeat test value was between 9 and 11. For anti-DENV IgG, a PU value of >22 and <18 were taken as positive and negative, respectively. Any value between 18 and 22 was considered equivocal and was tested with the same assay and considered negative if the repeat test value was between 18 and 22.

### Anti-SARS-CoV-2 IgG

The serum samples were tested for SARS-CoV-2 IgG using a commercial anti-SARS-CoV-2 spike-specific quantitative IgG (VITROS S-IgG) assay (Ortho VITROS Immunodiagnostics, New Jersy, USA) as per manufacturers' instructions. The assay kit detects anti-SARS-CoV-2 antibodies and is FDA-approved under Emergency Use Authorization. The measuring range (or linearity) of the kit was 2–200 BAU/ml. However, based on the limit of quantitation, values ≥17.8 BAU/ml were considered reactive, and otherwise non-reactive.

### Statistical analysis

DENV seroprevalence was estimated using the IgM and IgG levels and corrected using a pre-assessed sensitivity and specificity of the same tests. The corrected prevalence was calculated using the formula: (apparent prevalence + specificity—1)/(sensitivity + specificity—1). Force of infection (FOI) was calculated for estimating the seroprevalence in each district, using the WHO-FOI calculator, which assumes a constant FOI over time. The relationship between total population, population density, and mosquito clusters positive for DENV and DENV seropositivity in clinical samples was evaluated using binary logistic regression. The factors associated with DENV seropositivity as well as anti-DENV IgM/IgG levels were evaluated using binary and linear logistic regressions, respectively. Statistical analyses were performed using PRISM, ver.5.02 (GraphPad, San Diego, CA). Binary and linear regression was performed using SPSS, ver.20 (IBM, Armonk, NY), Two-tailed P<0.05 was considered significant, and P<0.05, P<0.01, and P<0.001 were marked as *, ** and ***, respectively.

## Results

### Spatial distribution of mosquito vectors in Tamil Nadu during 2017–24

To analyze the distribution of DENV-positive mosquitoes, the year-wise surveillance data of the State Public Health Laboratory, Chennai, and the Institute of Vector Control and Zoonoses, Hosur, Department of Public Health, Tamil Nadu between January 2016 and April 2024 were used for the comparison. The highest number of DENV-positive mosquito pools was observed during 2019, with 1440 of 3383 mosquito pools (42.6%). Though there was a two- to five-fold increase in the number of mosquito pools tested in subsequent years, there was a sudden decline during the SARS-CoV-2 pandemic with 8% positivity and a decreasing trend in DENV-positivity until mid-2024. In succeeding years from 2020 until April 2024, the DENV-positivity remained 3–8% (**Fig 1**).

### DENV infestation rate in mosquito pools collected from across the districts of Tamil Nadu, India

Next, we analysed the concurrent seroprevalence of dengue and SARS-CoV-2 in 2021. We noticed a surge in the global COVID-19 burden when mass vaccination programs were rolled out by the Government of India. Of a total of 9764 Aedes mosquito pools tested, 387 (3.96%)

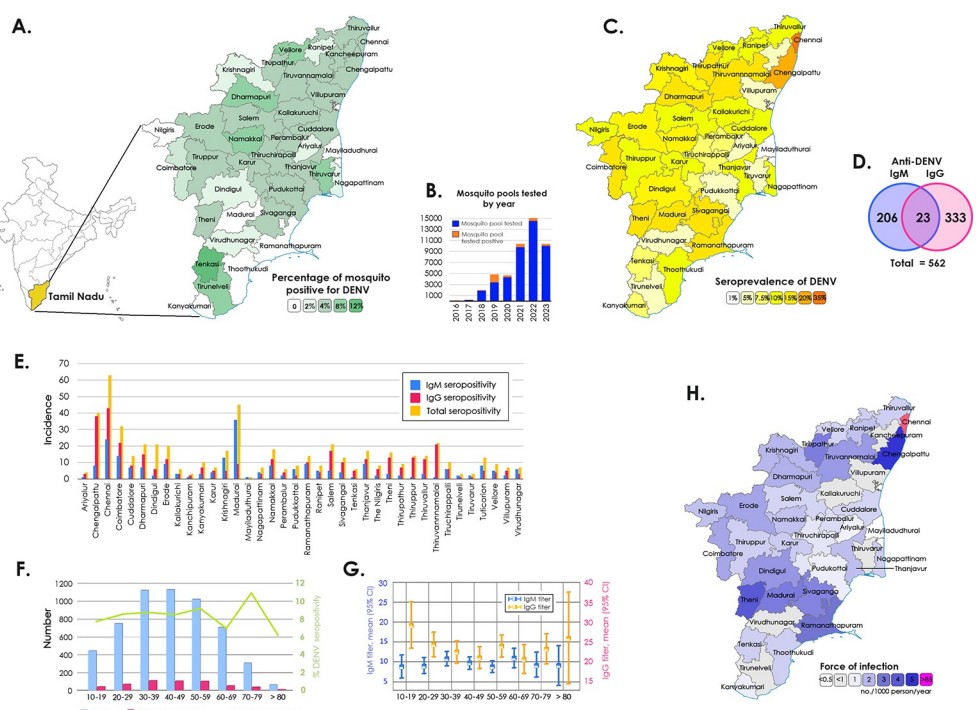

**Fig 1.** **A)** Spatial distribution of DENV-positive mosquito pools in Tamil Nadu **B)** Mosquito pools tested DENV positive by year **C)** Spatial distribution of seroprevalence of anti-DENV **D)** Number of seropositivities for anti-DENV IgM and IgG **E)** Distribution of DENV-seropositivity by districts **F-G)** Distribution of DENV-seropositivity across different age groups **H)** Estimation of the force of infection (FOI) of DENV. The country/region border shape was hand-drawn based on the base map layer as a template available for free from the Survey of India, Ministry of Science and Technology, Government of India (https://onlinemaps.surveyofindia.gov.in/Home.aspx).

tested positive for DENV (**Fig 1A and 1B**). A decline in the distribution of DENV-infected Aedes mosquitoes was observed during the survey period compared to previous years. Of the 38 districts, Madurai recorded the highest number (n = 644) of mosquito pools although the number of DENV-positive mosquito pools was highest in Tenkasi (11.1%) followed by Tirunelveli (8.07%) and Dharmapuri districts (7.89%). All the seven vector pools of the Nilgiris turned negative for DENV (**Fig 1A and 1B**). The district-wise distribution of DENV in mosquito pools is presented in the **S1 Table**.

## State-wide anti-DENV IgM and IgG seroprevalence in December 2021

Of the 5577 serum samples collected from 186 clusters, 230 samples (4.12%) were positive for anti-DENV IgM whereas 360 (6.4%) were positive for anti-DENV IgG. The highest seroprevalence of anti-DENV IgG was reported in Chennai with 24% while Madurai and Chennai districts recorded the highest (13%) seroprevalence of anti-DENV IgM (**Fig 1C–1E**). The age of the recruited population ranged from 10–96 years with a median of 43.6 years (**Table 1**). A high anti-DENV IgM positivity was observed among patients with 30–39 years (n = 51; 4.53%), 40–49 years (n = 52; 4.59%) and 80–89 years (n = 3; 4.84%) of age. Anti-DENV IgG positivity was higher among patients with 10–19 years (n = 31; 7.01%), 20–29 years (n = 54; 7.16%) and 70–79 years (n = 28; 8.95%) of age. All individuals aged between 90 and 99 years (n = 7) were negative for both DENV IgM and IgG. The IgM and IgG levels among males were 4.85% and 5.97%, whereas it was 3.57% and 6.81% among females, respectively. Anti-DENV IgM and IgG seroprevalence showed no significant difference among different age groups

**Table 1. Socio-demographic, clinical and serological characteristics of the study participants.**

| | |
|---|---|
| **Total number of participants,** *n* | 5577 |
| **Age,** year; *median (IQR)* | 43 (31–56) |
| **Gender,** male; *n (%)* | 2329 (41.8%) |
| **SARS-CoV-2 vaccination status,** yes; *n (%)* | 4654 (83.4%) |
| AZD1222; *n (%)* | 4249 (76.2%) |
| BBV152, *n (%)* | 395 (7.1%) |
| Others, *n (%)* | 10 (0.2%) |
| **History of SARS-CoV-2 infection;** *n (%)* | 166 (3%) |
| **Hospital admission,** *n (%)* | 77 (1.4%) |
| **SARS-CoV-2 IgG positivity,** *n (%)* | 4868 (87.3%) |
| **SARS-CoV-2 IgG titer,** *median (IQR)* | 200 (80.1–200) |
| **History of DENV infection;** *n (%)* | 15 (0.3%) |
| **Anti-DENV IgM positivity;** *n (%)* | 229 (4.1%) |
| **Anti-DENV IgG positivity;** *n (%)* | 356 (6.4%) |
| **Anti-DENV IgM titer;** *median (IQR)* | 2.5 (1.53–4.10) |
| **Anti-DENV IgG titer;** *median (IQR)* | 5.10 (2.42–9.54) |

All categorical variable reported as numbers (n) and percentages (%), and continuous variables reported as median, IQR.

(**Fig 1F and 1G**) and between the two genders. As the number of participants from the transgender community was low (n = 4), no analyses could be performed. The association of patients' domiciliary status (rural and urban) with total DENV seropositivity was highly significant. The seroprevalence of DENV in 38 districts of Tamil Nadu and the FOI (**Fig 1H**) in each district are listed in the **S2 Table** (**A-B**). Overall, the IgG seroprevalence and DENV-positivity in mosquito pools showed low (6.45% and 3.96%, respectively) during the study tenure.

## Relationship between anti-DENV seroprevalence and DENV positivity in mosquito pools

The correlation between seroprevalence (IgM/IgG/total) and DENV-positive mosquito pools was investigated using a Spearman correlation (**Fig 2A–2C**). The association significantly correlated with IgM (R = 0.305, 95% CI = 0.004–0.603; P = 0.04) and total seropositivity (R = 0.579, 95% CI = 0.026–1.132; P = 0.04), but not with IgG. The total DENV positivity was compared with DENV-positive mosquito clusters and DENV-FOI, which revealed no significance, although was evident with DENV-seropositivity (**Fig 2D–2F**). We also observed that DENV seroprevalence correlated significantly with factors such as total population (OR = 43.1, 95% CI = 31.1–55.2; P<0.001), percentage of mosquito infected by DENV (OR = 2.53, 95% CI = 0.47–4.59; P<0.01) and domiciliary status of participants (rural: OR = 18.3, 95% CI = 11.6–25.1; P<0.001; urban: OR = 28.9, 95% CI = 13.3–44.6; P<0.001) with increased odds (**Fig 2G**). The district-wise population-based seropositivity for SARS-CoV-2 IgG showed a high positivity rate ranging from 78–97% with a mean titre of 149.56 IU/ml (95% CI = 145.84–153.28IU/ml) (**Fig 3A**). The number of samples positive for IgG in each district is listed in the **S3 Table**.

The association of seroprevalence (IgM/IgG/total) with DENV-positive mosquito pools was investigated using a simple linear regression model with a 95% CI of the slope (**Fig 2A–2C**). The association was significantly correlated with IgM and total seropositivity, but not with IgG. The total DENV positivity was compared with DENV-positive mosquito clusters and

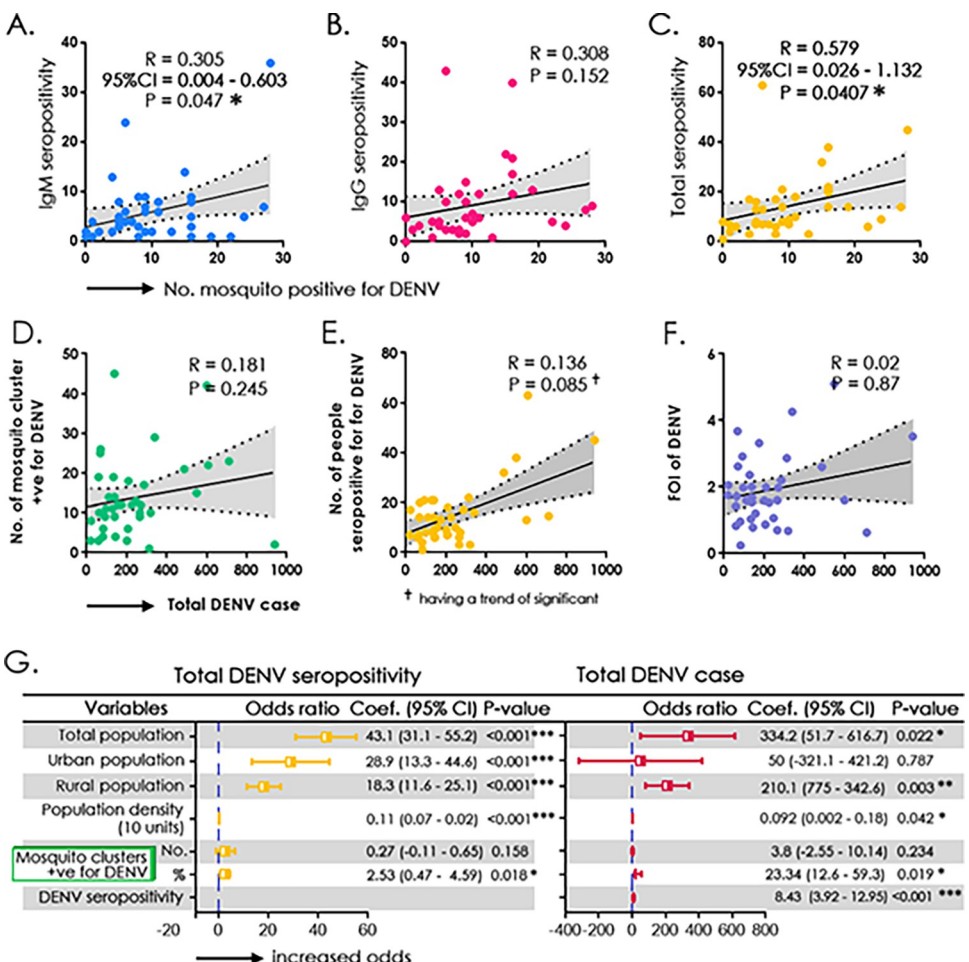

**Fig 2.** Correlation between a cluster of DENV-positive mosquito pools with A) DENV-IgM seropositivity **B)** DENV-IgG seropositivity **C)** Total DENV-seropositivity—Correlation between total anti-DENV IgM positive cases with: **D)** Number of clusters of DENV-infested mosquito pools, **E)** Number of individuals seropositive for DENV, and **F)** Force of infection of DENV **G)** Simple binary regression model assessing the relationship between total population, population density and mosquito clusters positive for DENV with total (IgG and IgM) seropositivity and anti-DENV IgM positive cases.

DENV-FOI, which revealed no significance, although was evident with DENV-seropositivity (**Fig 2D–2F**). We also observed that DENV seroprevalence correlated significantly with factors such as total population, DENV-infested mosquito clusters and domiciliary status of participants (rural/urban) with increased odds (**Fig 2G**). The district-wise population-based seropositivity for SARS-CoV-2 IgG showed a high titre ranging from 78–97% with a mean titre of 167 IU/ml (**Fig 3A**). The number of samples positive for IgG in each district is listed in the **S3 Table**.

## Analysis of clinico-demographic factors associated with DENV seroprevalence and anti-SARS-CoV-2 antibodies

Of the 5577 samples tested, 88.97% were reactive to SARS-CoV-2 IgG. Other factors including age, sex, vaccination status, type of vaccine administered and anti-SARS-CoV-2 antibodies were strongly associated with dengue seropositivity. While the levels of anti-SARS-CoV-2 correlated significantly with dengue seropositivity, vaccination status correlated similarly with anti-DENV IgM (**Fig 3C**). The association of two different types of SARS-CoV-2 vaccines, viz.,

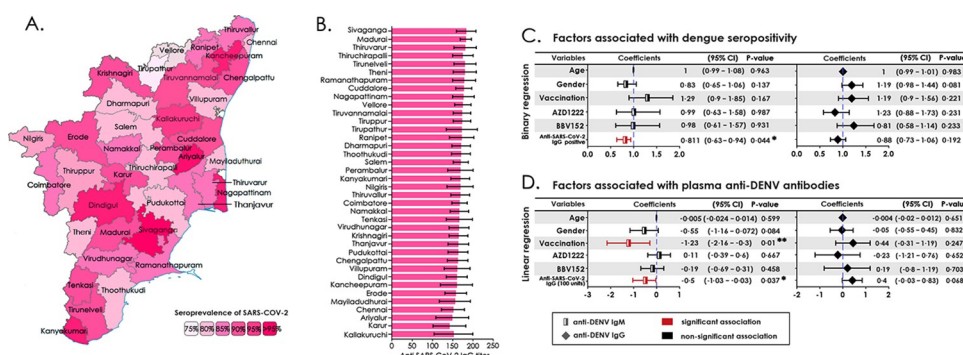

**Fig 3. A)** District-wise distribution of anti-SARS-CoV-2 seropositivity in Tamil Nadu **B)** Average levels of anti-SARS-CoV-2 IgG titer **C)** Binary regression model assessing the factors associated with the anti-DENV IgM and IgG seropositivity **D)** Linear regression model assessing the factors associated with the level of anti-DENV IgM and IgG. The country/region border shape was hand-drawn based on the base map layer as a template available for free from the Survey of India, Ministry of Science and Technology, Government of India (https://onlinemaps.surveyofindia.gov.in/Home.aspx).

BBV152 and AZD1222 with either anti-DENV IgM or IgG positivity did not reveal any significant difference (**Fig 3D**). The comparison of variables like total population, population density and number of DENV-positive mosquito clusters with DENV positivity is presented in **S4A and S4B Table**. The comparison of variables viz., age, gender, vaccination status and type of vaccine administered with DENV seropositivity is presented in **S5A and S5B Table**.

## Discussion

The district-wise distribution of DENV-positive mosquito pools and seroprevalence of anti-DENV and anti-SARS-CoV-2 antibodies were studied between December 2021 and March 2023. The anti-IgG, and anti-IgM levels as well as DENV-positivity in mosquito pools were low (6.45%, 3.9% and 3.96%, respectively). The mosquito pools positive for DENV correlated with anti-DENV IgM and total antibody levels, but not with IgG. Factors including the total population, DENV-infested mosquito clusters and domiciliary status of participants (rural/urban) showed a significant correlation with anti-DENV seroprevalence. Factors such as age, sex, vaccination status, type of vaccine administered and anti-SARS-CoV-2 were strongly associated with dengue seropositivity. The anti-SARS-CoV-2 IgG levels correlated significantly with dengue seropositivity, and, vaccination status correlated with anti-DENV IgM levels.

The burden of dengue fever and FOI poses considerable public health challenges to global health. However, the incidence is often underreported as most of the cases remain asymptomatic or misdiagnosed. The WHO data on global burden recorded the highest number of dengue cases (>6 million) and deaths (>7300) in 2023. DENV, being an arbovirus, has an ineludible link between human mobility, anthropogenic, and ecological factors [10]. Climate change and the spatio-temporal distribution of vectors due to the *El Niño* cycle, urbanization, population density and human mobility patterns represent key risk factors driving viral transmission and incidence rates [22,23]. In countries like India, despite the implementation of systematic vector surveillance/control programs and access to specific diagnostic tools, increased incidence of dengue fever poses a significant socio-economic burden [24]. An effective and all-inclusive vector and disease control program must therefore include serological, molecular and entomological surveillance for real-time monitoring of DENV circulation.

Both DENV and SARS-CoV-2 are associated with a high risk of severe disease and mortality rate. SARS-CoV-2 evolved into a pandemic in 2020 and thus far recorded >775.52 million

cases and 7.05 million deaths until 31 May 2024. After the first vaccine was rolled out in mid-December 2020, 5.5 billion doses have been administered globally. At least 56% of the global population is vaccinated at least with a complete primary series of COVID-19 vaccines and 28% of the population is vaccinated with at least one booster dose (World Health Statistics, WHO; available at www.who.int/data; last accessed on 01 June 2024). Immunization with highly effective and safe vaccines that produced a high titre of nAb reduced the disease burden significantly. However, emerging infections due to the circulating variants of concern with increased transmissibility and severity still pose a serious threat to global health. The vaccine-derived neutralizing antibodies did not offer cross-reactivity against the emerging new variants due to immune evasiveness influencing low transmission [25,26]. Serological cross-reactivity of anti-SARS-CoV-2 with dengue and Zika viruses, especially in DENV-endemic countries has been demonstrated previously [27].

The two viruses, despite having different routes of entry into their host, have similar pathogenesis, overlapping clinical presentations posing diagnostic predicaments and patient management. In addition, the antigenic similarities, heteroserotypic infection, and varying immunogenicity of the four DENV serotypes largely remain ambiguous. Hence, it is necessary to comprehensively analyze both the seroprevalence of DENV and SARS-CoV-2 at the community level to address the conundrum. In our study, the anti-DENV IgM and IgG titres were correlated with SARS-CoV-2 IgG in the community along with the virus screening in mosquitoes during December 2021. This was when both the cases and deaths declined upon vaccination in the timeline of the COVID-19 pandemic. The study was conducted across the state of Tamil Nadu, covering 38 administrative districts that had a population of 72 million.

The co-infection of DENV and SARS-CoV-2 further affects prognosis with increased mortality compared to either infection [28]. Convincing evidence of reduced dengue disease transmission was attributed to public health and social measures during the COVID-19 pandemic [10]. Reports of increased immature stages of Aedes mosquitoes due to COVID-19 lockdowns and subsequently interrupted larval control activities were expected to increase intra-household vector exposure and virus transmission [29]. A few countries reported increased dengue incidence during lockdowns [10,30], but a decline in cases was observed in a vast majority of countries [16].

India is one of the five highly endemic countries for dengue disease despite improved case management with a reduction in case-fatality rate to <0.5%. The Southeast Asian countries have witnessed a 46% upsurge in dengue cases between 2015 and 2019. A recent cross-sectional population-based serosurvey indicated 48.7% seroprevalence in India with the southern part of India which covered five states including Tamil Nadu recording the highest (77%). This indicated a high level of dengue transmission and geographical heterogeneity in the community during the pre-COVID-19 times [31]. A recent study reported a whopping 44.1% decrease in dengue across many dengue endemic regions, beginning in March 2020 (2.2 million cases in 2020 versus 4.08 million in 2019) [10]. In Tamil Nadu, a downward trend was observed in dengue-positivity from 8527 cases in 2019 to 2410 in 2020 followed by 6039 cases in 2021 and 6430 in 2022. In 2023, we recorded 4524 cases until September 2023 followed by a sudden spike in dengue cases during the post-monsoon season (October to December) with a total of 10570 cases. This could be attributed to related changes in social activities before and after the pandemic, cross-reactive serological tests with SARS-CoV-2 or increased and improved testing capacity of global laboratories.

Several limitations encountered in previous studies were addressed by including both IgM and IgG assays, testing of large clusters, and proportionate number of mosquito pools in our study design. We showed a district-wise distribution of DENV-positive mosquito pools ranging from 1–7.3% with an average of 3.8%. Seroprevalence of IgM ranged from 0.6–13.3% with

an average of 4.12% and IgG prevalence ranged from 1–23.9% with an average of 6.45%. The anti-SARS-CoV-2 IgG prevalence was high ranging from 78.3–97.8%. Interestingly, the anti-SARS-CoV-2 IgG titers correlated with dengue seropositivity indicating possible cross-protection, which however is unclear. Our data on vaccination status correlating with anti-DENV IgM levels needs further substantiation. Our present study used IgM and IgG Capture ELISA to estimate the antibody titres. IgG Capture ELISA is considered to be more sensitive compared to indirect IgG ELISA. The cut-off value for the detection of IgG is set lower for indirect ELISA compared to capture ELISA. Indirect ELISA is used in the dengue seroprevalence cohort [32] as a measure of past infection and is observed as one of the limitations of the study. However, the use of capture ELISA has been reported to be valuable for seroprevalence in otherwise healthy populations and for the differentiation of primary and secondary dengue [33]. In India, where Dengue is endemic, identification of high antibody titres is warranted [34]. Hence, we preferred to use capture ELISA which could help us identify past infection as well as secondary infections with high sensitivity and specificity. Studies on serosurveillance of dengue have been reported using capture ELISA [35].

The low incidence of dengue during the COVID-19 pandemic due to misdiagnosis or underreporting cannot be ruled out. Hence, all the intense measures to curtail disease transmission should be ascertained. The canonical analysis of virus distribution in mosquitoes together with seroprevalence in the human population should be an integral part of public health measures to curb mosquito populations and pathogen transmission. Further, in addition to the inclusion of both IgM and IgG seroprevalence, molecular typing could add details to the circulating dengue serotype and disease severity in the population.

The current study indicated the potential role of pre-existing anti-SARS-CoV-2 likely against DENV. However, the study appears to suffer from certain pitfalls that included the inability to demonstrate (1) validation of serological findings using dengue PRNT assays; (2) correlation of low titer of anti-SARS-CoV-2 with high anti-DENV seroprevalence in the population to further convince our findings; (3) cross-reactivity between the two viruses and viral interference by immune cells using human cell lines; (4) deviations in the clinical outcome; (5) DENV serotyping of circulating viruses; and (6) the association of human mobility, public health, and social constraints. The inclusion of these factors will prove any association between DENV and SARS-CoV-2 infections, and if this was a one-time phenomenon or observed at every emergence of SARS-CoV-2 variants across the globe.

In our study, we observed ~3.9% of DENV-positive mosquito pools which corroborated with previous studies in India that reported 1–5% DENV infection rates in Aedes mosquitoes [36]. Being a principal vector-borne disease across peri-urban and rural India, entomological surveys, by default, are a regular component of surveillance programs to assess vector density, spatial distribution, and coinciding dengue transmission in vectors and humans. Such integrated mosquito-human sampling across a specific geographical area could help assess the spatial seroprevalence pattern and assist identification of hot-spot regions of vector circulation. Such regions could be closely monitored and implement appropriate vector control measures to prevent outbreaks. Inadequacy in addressing virus transmission systems has been incriminated with futile vector control programs [37]. Our study identified the ecological distribution of DENV-positive mosquito pools across Tamil Nadu.

The district-wise distribution of DENV-positive vectors and seroprevalence of DENV and SARS-CoV-2 was studied between December 2021 and March 2023. The IgG, IgM seroprevalence and DENV-positivity in mosquitos were low (6.45%, 3.9% and 3.96%, respectively). The mosquito DENV-positivity correlated with DENV-IgM and total antibody levels, but not with IgG. Factors including the total population, DENV-infested mosquito clusters and domiciliary status of participants (rural/urban) showed a significant correlation with DENV

seroprevalence. Factors such as age, sex, vaccination status, type of vaccine administered and anti-SARS-CoV-2 antibodies were strongly associated with dengue seropositivity. The anti-SARS-CoV-2 IgG levels correlated significantly with dengue seropositivity, and COVID-19 vaccination status correlated similarly with anti-DENV IgM levels. This could be due to possible interference of SARS-CoV-2 antibodies to DENV infection. A study also showed an association of anti-SARS-CoV-2 antibodies with reduced DENV seroprevalence with higher odds, specifically in the rural population [38]. Another study ruled out any possible cross-reactivity between the two viruses [39]. This indicates the possible implications of SARS-CoV-2 infection and subsequent vaccination on reduced DENV positivity in mosquitoes and seroprevalence in humans. Viral interference is a state of momentary immunity caused by a viral infection that precludes replication by other viruses [40]. A heterologous virus interference can be a possible explanation for the reduced prevalence of dengue due to anti-SARS-CoV-2 antibodies. Besides co-infection, the interactions between these two viruses have not been adequately studied. The understanding of the intrinsic viral interference between SARS-CoV-2 and DENV could shed more insights for better public health interventions and vaccine developments.

## Conclusions

Dengue fever and SARS-CoV-2 continue to remain major global public health concerns, predominantly in the tropical world where dengue case incidence is exponentially increasing annually, and there is an ongoing geographical expansion of transmission areas and cocirculation of multiple DENV serotypes. It is of paramount importance to establish laboratory-based sentinel surveillance with coordinated entomological and molecular surveillance for early diagnosis, prevention, and control of arboviral infections.

## Supporting information

**S1 Checklist. Inclusivity in global research.**
(DOCX)

**S1 Table. District-wise DENV positivity in mosquito pools in 2023.**
(PDF)

**S2 Table. District-wise distribution of DENV seropositivity.**
(PDF)

**S3 Table. District-wise distribution of SARS-CoV-2 seropositivity.**
(PDF)

**S4 Table. a.** Mosquito pools positive for DENV and DENV seropositivity in the community. **b.** Mosquito positivity for DENV, and DENV positive cases.
(PDF)

**S5 Table. a.** Factors associated with DENV seropositivity. **b.** Factors associated with DENV IgM and IgG levels.
(PDF)

## Author Contributions

**Conceptualization:** Sivaprakasam T. Selvavinayagam, Sathish Sankar, Yean K. Yong, Anavarathan Somasundaram, Manickam Malathi, Ramendra P. Pandey, Amudhan Murugesan, Siddappa N. Byrareddy, Vijayakumar Velu, Marie Larsson, Esaki M. Shankar, Sivadoss Raju.

**Data curation:** Sathish Sankar, Yean K. Yong, Abdul R. Anshad, Samudi Chandramathi, Anavarathan Somasundaram, Sampath Palani, Parthipan Kumarasamy, Roshini Azhaguvel, Ajith B. Kumar, Sudharshini Subramaniam, Manickam Malathi, Venkatachalam Vijayalakshmi, Manivannan Rajeshkumar, Anandhazhvar Kumaresan, Nagarajan Muruganandam, Amudhan Murugesan, Sivadoss Raju.

**Formal analysis:** Sathish Sankar, Yean K. Yong, Samudi Chandramathi, Roshini Azhaguvel, Sudharshini Subramaniam, Manickam Malathi, Venkatachalam Vijayalakshmi, Ramendra P. Pandey, Nagarajan Muruganandam, Meganathan Kannan, Siddappa N. Byrareddy, Aditya P. Dash, Vijayakumar Velu, Esaki M. Shankar, Sivadoss Raju.

**Funding acquisition:** Sivaprakasam T. Selvavinayagam, Vijayakumar Velu, Marie Larsson, Sivadoss Raju.

**Investigation:** Sivaprakasam T. Selvavinayagam, Abdul R. Anshad, Anavarathan Somasundaram, Sampath Palani, Parthipan Kumarasamy, Roshini Azhaguvel, Venkatachalam Vijayalakshmi, Anandhazhvar Kumaresan, Nagarajan Muruganandam, Natarajan Gopalan, Meganathan Kannan, Pachamuthu Balakrishnan, Marie Larsson, Esaki M. Shankar, Sivadoss Raju.

**Methodology:** Sathish Sankar, Abdul R. Anshad, Parthipan Kumarasamy, Roshini Azhaguvel, Ajith B. Kumar, Sudharshini Subramaniam, Manickam Malathi, Manivannan Rajeshkumar, Anandhazhvar Kumaresan, Meganathan Kannan, Amudhan Murugesan, Aditya P. Dash.

**Project administration:** Sampath Palani, Parthipan Kumarasamy, Sudharshini Subramaniam, Venkatachalam Vijayalakshmi, Aditya P. Dash, Vijayakumar Velu, Sivadoss Raju.

**Resources:** Sathish Sankar, Yean K. Yong, Anavarathan Somasundaram, Sampath Palani, Parthipan Kumarasamy, Roshini Azhaguvel, Ajith B. Kumar, Sudharshini Subramaniam, Manickam Malathi, Venkatachalam Vijayalakshmi, Manivannan Rajeshkumar, Anandhazhvar Kumaresan, Ramendra P. Pandey, Nagarajan Muruganandam, Amudhan Murugesan, Pachamuthu Balakrishnan, Siddappa N. Byrareddy, Aditya P. Dash, Vijayakumar Velu, Marie Larsson, Esaki M. Shankar, Sivadoss Raju.

**Software:** Sathish Sankar, Yean K. Yong, Samudi Chandramathi, Sudharshini Subramaniam, Manickam Malathi, Venkatachalam Vijayalakshmi, Anandhazhvar Kumaresan, Ramendra P. Pandey, Nagarajan Muruganandam, Natarajan Gopalan, Amudhan Murugesan, Pachamuthu Balakrishnan, Siddappa N. Byrareddy, Aditya P. Dash, Vijayakumar Velu, Marie Larsson, Esaki M. Shankar.

**Supervision:** Sivaprakasam T. Selvavinayagam, Ajith B. Kumar, Natarajan Gopalan, Pachamuthu Balakrishnan, Marie Larsson, Esaki M. Shankar, Sivadoss Raju.

**Validation:** Sivaprakasam T. Selvavinayagam, Sathish Sankar, Yean K. Yong, Samudi Chandramathi, Anavarathan Somasundaram, Sampath Palani, Parthipan Kumarasamy, Roshini Azhaguvel, Ajith B. Kumar, Venkatachalam Vijayalakshmi, Anandhazhvar Kumaresan, Ramendra P. Pandey, Nagarajan Muruganandam, Natarajan Gopalan, Meganathan Kannan, Amudhan Murugesan, Pachamuthu Balakrishnan, Siddappa N. Byrareddy, Aditya P. Dash, Vijayakumar Velu, Marie Larsson, Esaki M. Shankar, Sivadoss Raju.

**Visualization:** Sathish Sankar, Manivannan Rajeshkumar, Anandhazhvar Kumaresan, Nagarajan Muruganandam, Natarajan Gopalan, Meganathan Kannan, Amudhan Murugesan, Vijayakumar Velu, Marie Larsson.

**Writing – original draft:** Sivaprakasam T. Selvavinayagam, Sathish Sankar, Yean K. Yong, Samudi Chandramathi, Ramendra P. Pandey, Natarajan Gopalan, Vijayakumar Velu, Esaki M. Shankar, Sivadoss Raju.

**Writing – review & editing:** Yean K. Yong, Manickam Malathi, Venkatachalam Vijayalakshmi, Nagarajan Muruganandam, Pachamuthu Balakrishnan, Siddappa N. Byrareddy, Aditya P. Dash, Vijayakumar Velu, Marie Larsson, Esaki M. Shankar, Sivadoss Raju.

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
