## [Decision Letter · Decision Letter 0]

5 Jul 2024

PGPH-D-24-01370

Serosurveillance of dengue infection and correlation with aedine mosquito pools for dengue virus positivity during the COVID-19 pandemic in Tamil Nadu, India – A state-wide cross-sectional cluster-randomized community-based study

Dear Dr.Shankar,

Thank you for submitting your manuscript to PLOS Global Public Health. After careful consideration, we feel that it has merit but does not fully meet PLOS Global Public Health’s publication criteria as it currently stands. Therefore, we invite you to submit a revised version of the manuscript that addresses the points raised during the review process.

We look forward to receiving your revised manuscript.

Kind regards,

André Machado Siqueira, M.D., MSc, Ph.D

Academic Editor

Journal Requirements:

Additional Editor Comments (if provided):

Reviewers' comments:

Reviewer's Responses to Questions

**Comments to the Author**

1. Does this manuscript meet PLOS Global Public Health’s publication criteria? Is the manuscript technically sound, and do the data support the conclusions? The manuscript must describe methodologically and ethically rigorous research with conclusions that are appropriately drawn based on the data presented.

Reviewer #1: Partly

Reviewer #2: Partly

Reviewer #3: Partly

2. Has the statistical analysis been performed appropriately and rigorously?

Reviewer #1: No

Reviewer #2: I don't know

Reviewer #3: I don't know

3. Have the authors made all data underlying the findings in their manuscript fully available (please refer to the Data Availability Statement at the start of the manuscript PDF file)?

Reviewer #1: No

Reviewer #2: Yes

Reviewer #3: Yes

4. Is the manuscript presented in an intelligible fashion and written in standard English?

Reviewer #1: Yes

Reviewer #2: No

Reviewer #3: Yes

5. Review Comments to the Author

Reviewer #1: The authors utilized a randomized cluster sampling community-based survey to assess anti-dengue IgM and IgG, as well as SARS-CoV-2 IgG seroprevalence across all 38 districts of Tamil Nadu, India. Overall, the manuscript needs major and minor editions. Furthermore, the lack of information about the pre-analytical phase and sample preservation hinders addressing significant issues such as confounding, selection, and information biases, which are prevalent in seroprevalence studies. The researchers observed seroprevalence levels significantly lower than those reported previously and their own seroprevalence estimates used for sample size calculation. Therefore, it remains unclear what reasons could account for these discrepancies aside from issues during the laboratory pre-analytical phase and the absence of controls during sample processing. This is concerning because, without any data on the reliability of the diagnostic procedure, reviewers are unable to assess the study's internal validity. This dearth of information is critical in seroprevalence studies as it can lead to biased results. Prior to publication, the authors should provide additional information on data quality control measures to enable reviewers to evaluate the study's internal validity. Therefore, I cannot recommend the manuscript for publication at this time.

MAJOR COMMENTS

Overall

• Based on the review of the manuscript, it is evident that further information regarding the pre-analytical phase and sample preservation is essential to assess the study's internal validity accurately. The lack of details on these crucial aspects raises concerns about potential biases that could impact the study's outcomes.

• The title is misleading, and the authors did not provide any evidence that match the study title.

Study Design

• Please clarify when the samples were collected and when they were processed to understand the preservation period. Furthermore, if possible, describe whether did you store controls to assess the samples preservation and the results of these assessments. If not, reflect this as a limitation in the discussion section.

• Please provide further information on the pre-analytical phase, including sample collection procedures and handling to prevent biases. This additional information is crucial for assessing the reliability and accuracy of the study results.

• Please explain in detail the measures taken to preserve serum samples and the duration between sample collection and processing.

• Please describe the steps taken to avoid antibody degradation prior to testing to ensure the accuracy of seroprevalence estimates.

• A word of caution about correlation is not causation unless you perform further analysis, so I will strongly recommend just limit your study to describe the trends and assessed using mathematical models.

Statistical Analysis

• Please describe the statistical methods you used to test your study hypothesis about the “correlation” between the dengue seroprevalence and the aedine mosquito pools for dengue virus positivity during the COVID-19 pandemic.

• Please review your method for correcting your prevalence estimates and explain why you are adding the “apparent prevalence” to the “specificity”.

Results

• Please double check your study period. You present different period of interest across the results that do not match the study methods.

• Please consider including a trend analysis

Discussion

• Please be use a more conservative and objective approach to present your study results and conclusions and use the first paragraph of your discussion to present the main study results.

• Please discuss your results instead about discussing unrelated issues.

• Please avoid misleading discussions about co-infection of DENV and SARS-CoV-2 because your study does not present any evidence about or against this issue.

MINOR COMMENTS

• Please include the confidence interval of your estimates in your results and graphics.

• Seek a secondary review from a statistician with experience in seroprevalence studies to review your data analysis.

Reviewer #2: 1. General:

- manuscript not line-numbered so that it is difficult to refer when reviewing.

- Authors should clearly refer to Figures in detail, including mentioning the alphabet on the Figure in the text (e.g. Fig 1A, Fig 1B, etc).

2. Abstract:

- please state clearly the aim of the study

- what is the purpose of mentioning the binary and linear regression method in the Abstract?

- is there conclusion in the Abstract?

- please consider to use structured abstract

3. Method:

- what type of Panbio IgM/IgG ELISA? Capture ELISA? Capture ELISA tends to use in acute infection/patients. The use of this ELISA in healthy subjects may underestimate the IgG seropositivity results. Author should use Indirect ELISA.

- RT-PCR detection of DENV in mosquito is very tricky due to the presence of PCR inhibitors in the mosquito lysates. Does the method has been validated in mosquito samples? How to ensure that the RT-PCR does not falsely detect the Insect-Specific Flaviviruses that are very commonly presence in mosquito samples?

Results:

- what is the correlation of "the concurrent seroprevalence of dengue and SARS-CoV-2 in 2021" with mosquito samples assay? (it is difficult to refer the sentence when the manuscript is not line numbered!!!)

- why NS1 is not used? Why only IgG/IgM>

- the very high positivity rates of DENV in mosquito pools is somewhat doubtful. Typically, very low positivity rates is commonly observed when detecting DENV in mosquito samples. Authors must provide strong evidence that the high positivity rates is not caused by false-positive detection of Insect-specific Flaviviruses. It is strongly suggested that the authors isolate the virus or sequence the genome of the virus to proof that the detected DENV are truly from DENV.

- what are the serotype of the DENV detected in mosquitoes?

- were positive controls (mosquito positive for DENV) used in the experiments?

Discussion:

- COVID-19 pandemic has passed and many publication on dengue and COVID-19 are available. What is the novelty of this study?

- Discussion is a bit more general and not specifically discuss the data. Authors should put more attention on the findings, its novelty, and the benefit of the finding.

Conclusion:

- it's not summarizing the finding but instead only stating about the importance of dengue and COVID-19.

- so what is the main finding in brief?

- what is the importance of this study?

Reviewer #3: 1. Manuscript describes some interesting, but very intriguing findings on the drastic reduction of Dengue seroprevalence in Tamil Nadu, India, during the COVID Pandemic. Previous studies have reported a seroprevalence of about 70.0% in South India.

2. The reasons attributed may not be conclusive. Another drawback of the study is that they did not cross check their findings using a random Dengue PRNT. The present reviewer could suspect less sensitivity and specificity of the Pan bio-ELISA IgM and IgG kit used for detecting the DENV seroprevalence, owing to the interference of SARS COV-2 virus infection (?) would have been a crucial factor in this drastic reduction of seropositivity on detecting Dengue virus antigen.

3. However, Authors try to postulate many aspects and attributes to this drastic reduction in DENV seroprevalence to many factors, the major one being SARS COV-2 virus infection, without proof for any proposed concepts.

4. Authors do not provide any references to their methodologies used, in the study.

5. Towards sampling of the population, authors calculate the sample size using a dengue seroprevalence rate to be 76.9%. The outcome results on the seroprevalence are 4.12% for IgM and 6.4% for IgG. An expert statistician has to verify, whether this sampling procedure had been correct, in the study!

6. Some minor grammatical errors are there in the write up. May be corrected.

7. Introduction line 2. Aedes aegypti and Aedes albopictus should be written fully, instead of abbreviating those as Ae. aegypti and Ae. albopictus, when using species names for the first time in a write-up.

6. PLOS authors have the option to publish the peer review history of their article (what does this mean?). If published, this will include your full peer review and any attached files.

**Do you want your identity to be public for this peer review?** For information about this choice, including consent withdrawal, please see our Privacy Policy.

Reviewer #1: **Yes: **Antonio M Quispe

Reviewer #2: No

Reviewer #3: No

---

## [Decision Letter · Decision Letter 1]

16 Sep 2024

PGPH-D-24-01370R1

Attrition in serum anti-DENV antibodies correlates with high anti-SARS-CoV-2 IgG levels and low DENV positivity in mosquito vectors - Findings from a state-wide cluster-randomized community-based study in Tamil Nadu, India

Dear Dr. Shankar,

Thank you for submitting your manuscript to PLOS Global Public Health. After careful consideration, we feel that it has merit but does not fully meet PLOS Global Public Health’s publication criteria as it currently stands. Therefore, we invite you to submit a revised version of the manuscript that addresses the points raised during the review process.

We look forward to receiving your revised manuscript.

Kind regards,

André Machado Siqueira, M.D., MSc, Ph.D

Academic Editor

Additional Editor Comments (if provided):

Reviewers' comments:

Reviewer's Responses to Questions

**Comments to the Author**

1. If the authors have adequately addressed your comments raised in a previous round of review and you feel that this manuscript is now acceptable for publication, you may indicate that here to bypass the “Comments to the Author” section, enter your conflict of interest statement in the “Confidential to Editor” section, and submit your "Accept" recommendation.

Reviewer #1: All comments have been addressed

Reviewer #2: (No Response)

Reviewer #3: (No Response)

2. Does this manuscript meet PLOS Global Public Health’s publication criteria? Is the manuscript technically sound, and do the data support the conclusions? The manuscript must describe methodologically and ethically rigorous research with conclusions that are appropriately drawn based on the data presented.

Reviewer #1: Yes

Reviewer #2: Partly

Reviewer #3: Partly

3. Has the statistical analysis been performed appropriately and rigorously?

Reviewer #1: Yes

Reviewer #2: I don't know

Reviewer #3: No

4. Have the authors made all data underlying the findings in their manuscript fully available (please refer to the Data Availability Statement at the start of the manuscript PDF file)?

Reviewer #1: Yes

Reviewer #2: No

Reviewer #3: Yes

5. Is the manuscript presented in an intelligible fashion and written in standard English?

Reviewer #1: Yes

Reviewer #2: Yes

Reviewer #3: No

6. Review Comments to the Author

Reviewer #1: Thank you for responding to the review appropriately.

Reviewer #2: While authors have answered some of the concerns from reviewers, some important comments are not satisfactorily answered. The following points are very important for the data integrity and without further action from authors, the manuscript is not appropriate for publication:

1. Authors do not agree to use indirect IgG ELISA which is more appropriate for sero-surveillance study.

2. Authors fail to prove that the very high positivity rates of DENV in mosquitoes is real and not because of non-specific amplification.

3. No serotype data is shown, as well as no genome sequence data to prove that the amplified DENV from mosquitoes are indeed from DENV, not from non-specific amplification/artefacts.

Reviewer #3: Authors have not addressed the comments carefully in the revised manuscript.

Still, I fear the conclusions arrived at IN THE MANUSCRIPT may put forth the misleading concept, that during covid pandemic Dengue incidence reduced drastically, without carrying out or citing any valid proof of concept scientifically.

1) The drastic reduction of dengue incidence recorded may be due to the interference of SARS COV-2 infection preventing immunological detection of Dengue positives, as I have commented earlier.

Kindly refer the publication by Cheng et al., 2022, "Antibodies against the SARS-CoV-2 S1-RBD cross-react with dengue virus and hinder dengue pathogenesis", Frontiers in Immunology, DOI: 10.3389/fimmu.2022.941923, which states that there is a clear interference of SARS COV-2 virus in the detection of Dengue cases by immunologically, as well as in the pathogenesis.

I think this reported complex interactions, would have been the major attribute for the reduced incidence of Dengue detected using ELISA kits during the pandemic.

Hence the conclusions could have been in these lines, possibly after carrying out a proof of concept study.

2) In case of selection of sample size also, there exist a clear error in the study. In the reply to comments you state that the sampling size was selected based on Murhekar et al, 2019, who comes out with an incidence of 76.9% for Dengue. But in your study which reports the incidence of Dengue to be about 6.4%. Thus the sample size you selected is an underrepresentation. You could have carried out a pre-test to estimate the sample size of the study.

3) Regarding xenomonitoring also, you have screened the samples by RT-PCR. You do not provide data on samples processed from different districts for years 2016-2023. In supplementary data you include only a table (Table S1) on the spatial distribution of mosquitoes sampled during 2023. Have you compared data through different years? Was the methodology of detection followed from 2016-2023 same?

You should have processed random samples using conventional PCR detection of CprM of E genetic markers of DENV and should have sequenced those to confirm the specificity and sensitivity of your detections.

Besides study have many limitations, a least a few could have been avoided to make study more scientific.

7. PLOS authors have the option to publish the peer review history of their article (what does this mean?). If published, this will include your full peer review and any attached files.

**Do you want your identity to be public for this peer review?** For information about this choice, including consent withdrawal, please see our Privacy Policy.

Reviewer #1: **Yes: **Antonio Marty Quispe

Reviewer #2: No

Reviewer #3: No

---

## [Decision Letter · Decision Letter 2]

4 Nov 2024

Attrition in serum anti-DENV antibodies correlates with high anti-SARS-CoV-2 IgG levels and low DENV positivity in mosquito vectors - Findings from a state-wide cluster-randomized community-based study in Tamil Nadu, India

PGPH-D-24-01370R2

Dear Dr Shankar,

We are pleased to inform you that your manuscript 'Attrition in serum anti-DENV antibodies correlates with high anti-SARS-CoV-2 IgG levels and low DENV positivity in mosquito vectors - Findings from a state-wide cluster-randomized community-based study in Tamil Nadu, India' has been provisionally accepted for publication in PLOS Global Public Health.

Best regards,

André Machado Siqueira, M.D., MSc, Ph.D

Academic Editor

Reviewer Comments (if any, and for reference):

Reviewer's Responses to Questions

**Comments to the Author**

1. If the authors have adequately addressed your comments raised in a previous round of review and you feel that this manuscript is now acceptable for publication, you may indicate that here to bypass the “Comments to the Author” section, enter your conflict of interest statement in the “Confidential to Editor” section, and submit your "Accept" recommendation.

Reviewer #3: All comments have been addressed

2. Does this manuscript meet PLOS Global Public Health’s publication criteria? Is the manuscript technically sound, and do the data support the conclusions? The manuscript must describe methodologically and ethically rigorous research with conclusions that are appropriately drawn based on the data presented.

Reviewer #3: Yes

3. Has the statistical analysis been performed appropriately and rigorously?

Reviewer #3: Yes

4. Have the authors made all data underlying the findings in their manuscript fully available (please refer to the Data Availability Statement at the start of the manuscript PDF file)?

Reviewer #3: Yes

5. Is the manuscript presented in an intelligible fashion and written in standard English?

Reviewer #3: Yes

6. Review Comments to the Author

Reviewer #3: Most of the comments provided had been addressed in the revised manuscript. The Discussions and conclusions are more scientific at present. You have also presented the limitations of the study.

7. PLOS authors have the option to publish the peer review history of their article (what does this mean?). If published, this will include your full peer review and any attached files.

**Do you want your identity to be public for this peer review?** For information about this choice, including consent withdrawal, please see our Privacy Policy.

Reviewer #3: **Yes: **Dr. N. Pradeep Kumar
